# Drug Treatment Direction Based on the Molecular Mechanism of Breast Cancer Brain Metastasis

**DOI:** 10.3390/ph18020262

**Published:** 2025-02-16

**Authors:** Yumin Zhang, Haotian Shang, Jiaxuan Zhang, Yizhi Jiang, Jiahao Li, Huihua Xiong, Tengfei Chao

**Affiliations:** 1Department of Oncology, Tongji Hospital, Tongji Medical College, Huazhong University of Science and Technology, Wuhan 430030, China; zhangmin200186@gmail.com (Y.Z.); m202276295@hust.edu.cn (H.S.); m202376699@hust.edu.cn (Y.J.); m202276284@hust.edu.cn (J.L.); 2Department of Radiology, Tongji Hospital, Tongji Medical College, Huazhong University of Science and Technology, Wuhan 430030, China; jiaxuanzhang@126.com

**Keywords:** BCBM, Her2, WNT pathway, NOTCH pathway, CDK4/6inhibitors, BBB, LRP-1, VEGF, nanotechnology

## Abstract

Today, breast cancer (BC) is the most frequently diagnosed malignancy and a leading cause of cancer-related deaths among women worldwide. Brain metastases (BMs) are a common complication among individuals with advanced breast cancer, significantly impacting both survival rates and the overall condition of life of patients. This review systematically analyzes the innovative approaches to drug treatment for breast cancer brain metastases (BCBMs), with particular emphasis placed on treatments targeting molecular mechanisms and signaling pathways and drug delivery strategies targeting the blood brain barrier (BBB). The article discusses various drugs that have demonstrated effectiveness against BCBM, featuring a mix of monoclonal antibodies, nimble small-molecule tyrosine kinase inhibitors (TKIs), and innovative antibody-drug conjugates (ADCs). This study of various drugs and techniques designed to boost the permeability of the BBB sheds light on how these innovations can improve the treatment of brain metastases. This review highlights the need to develop new therapies for BCBM and to optimize existing treatment strategies. With a deeper comprehension of the intricate molecular mechanisms and advances in drug delivery technology, it is expected that more effective personalized treatment options will become available in the future for patients with BCBM.

## 1. Introduction

BC is currently the most prevalent neoplasm within females of all ages worldwide. Of patients with metastatic breast cancer (MBC), roughly 15–30% develop brain metastases [1]. Over the last decade, median overall survival (OS) in patients with BCBMs has consistently been suboptimal, with only a small proportion surviving for 5 years or more [2]. BCBMs are swiftly emerging as a significant impediment to the survival of breast cancer patients. Previously, the treatment of survivors with BCBMs was based on localized treatments, like radiation therapy or surgery, but today, the utilization of drugs as systemic therapy is becoming increasingly important.

BMs develop for various reasons, like the molecular characteristics of cancer cells and signaling pathway alterations. Moreover, the BBB, as a special neurovascular unit, limits the penetration of antitumor drugs into the brain [3]. Many researchers have focused on these issues, particularly regarding anti-HER2 treatment and local treatment. In addition to focusing on anti-HER2 treatment, this review emphasizes other potential targets and studies clinical drugs, such as CDK4/6 inhibitors, Trop-2, PD-1/PD-L1, WNT and Notch pathway inhibitors. In addition, the role of BBB in brain metastasis in breast cancer is discussed, and methods for treating BCBM are explained through BBB delivery drugs, such as physical pathways to increase BBB permeability, the use of nanotechnology for drug delivery, and drug delivery routes targeting BBB endothelial cells. This review will discuss the therapeutic targeted drugs used for BCBM based on its molecular mechanisms and describe the role of the BBB in delivering drugs for treatment (Figure 1).

## 2. Mechanisms of Brain Metastasis

From primary site development to the colonization in the brain, multiple steps are required (Figure 1), including epithelial-mesenchymal transition (EMT), invasion, endocytosis, and extravasation [4]. Among these processes, EMT is a reversible procedure wherein mesenchymal characteristics are acquired by breast cancer cells (BCCs) [5]. Furthermore, mesenchymal cells are more metastatic and invasive than epithelial cells as they reduce the production of transmembrane E-calmodulin5 and increase the secretion of matrix metalloproteinase (MMP) and foster the breakdown of basement membrane proteins [6,7]. These traits mean that BCCs are more likely to enter the bloodstream. This allows BCCs to access the blood circulation system more easily so they may develop into circulating tumor cells (CTCs). BCCs’ entry into the blood circulatory system is referred to as endocytosis, and the mechanism that can be identified is the involvement of macrophages. Intravascular macrophages attract tumor cells by producing epidermal growth factor (EGF) and release vascular endothelial growth factor (VEGF) to encourage EC proliferation, leading to the disruption of junctions between ECs, resulting in increased vascular permeability, which facilitates the entry of BCCs into the circulatory system from the paracellular pathway across the vascular endothelium [8]. In addition, the activation of Notch signaling stimulates the migration of BCCs. The activation of Notch signaling prompts Notch ligands on the EC surface to bind to Notch receptors on the BCC surface, thereby stimulating BCC migration [9]. CTC extravasates and colonizes the brain with blood circulation, and there are three types of BCBM: parenchymal, leptomeningeal, and choroid plexus metastases. Choroid plexus metastases are rare, leptomeningeal metastases account for 8% of cases, and parenchymal metastases are the most commonly occurring kind of brain metastasis, comprising 78% of multiple brain metastases and 14% of isolated brain metastases [10].

The extravasation of CTC into the brain parenchyma requires crossing the BBB, formed by ECs, pericytes and astrocytes that collectively regulate the homeostasis of the central nervous system (CNS), which collectively maintains normal brain function, with ECs significantly influencing the invasive capacitise of cancer cells [11]. It has been found that CTCs migrate within the vasculature by rolling, adhering, and, finally, completing transendothelial migration (TEM). Endothelial selectin (E-selectin) on the surface of ECs binds to the E-selectin ligand, CD44, and mucin 1 on cancer cells to guide the rolling process [4]. Intercellular adhesion molecule-1 (ICAM-1) and vascular cell adhesion molecule-1 (VCAM-1) are overexpressed on the EC surface, with CTCs adhering to the ECs [12]. It was found that CTC mainly crosses the ECs through the paracellular pathway to complete TEM, and paracellular permeability is a major factor in TEM [13].

Paracellular permeability is predominantly regulated by the tight junctions (TJs) that exist between ECs. This regulation involves transmembrane proteins like occludins, claudins, and junctional adhesion molecules (JAMs), serving as essential components of TJs [10]. The adhesion of CTCs to the endothelium induces the destruction of these protein. Multiple mechanisms ultimately lead to TEM for CTC [14]. In addition, like endocytosis, VEGF is essential in exocytosis. The results of too high VEGF and angiopoietin-2 reduced the production of TJ proteins, disturbed intercellular junctions, and increased BBB permeability [4]. Moreover, astrocytes and pericytes preserve the functionality of the BBB. Astrocytes directly help ECs to grow and proliferate by secreting angiotensin 1 (ANG-1) and VEGF, thereby affecting the permeability of the BBB [15]. Pericytes are in close proximity to the EC and directly influence BBB permeability [16].

BCCs infiltrate and grow to form distant metastases, whereas glioblastoma (GBM) originates from glial cell tissue, grows infiltratively into the brain parenchyma, and almost never metastasizes [17]. However, both types of tumors must overcome the physical barriers provided by myoepithelial and astrocyte protrusions. In GBM, glioma cells are in direct contact with endothelial cells to destroy BBB. Additionally, in glioma, the existing vasculature is absorbed during initial tumorigenesis; however, subsequent tumor growth and hypoxia induce neoangiogenesis. This suggests the potential efficacy of VEGF inhibitors [15]. For intracerebral tumors, it is crucial to use drugs that can cross the BBB. Preclinical studies have shown that temozolomide can distribute across the BBB, blood-tumor barrier (BTB), and cerebrospinal fluid (CSF) in mice [18]. Currently, temozolomide is an important component of drug treatment for glioblastoma. However, its efficacy in breast cancer brain metastasis (BCBM) patients is limited [19], highlighting that different tumors exhibit varying BTB permeability.

Despite these differences, the overall treatment strategy remains the same: to deliver drugs across the BTB as effectively as possible to achieve therapeutic concentrations in the central nervous system (CNS).

## 3. Therapeutic Strategies Based on Molecular Mechanisms of BCBM

BC exhibits a higher propensity for the development of BM in advanced stages, and BM prognosis is usually poor. BM is particularly prevalent in the HER2-positive subtype of BC [20]. In recent years, many studies have identified various genes and abnormal signaling pathways mediating BCBM. These findings have advanced our comprehension of BM and provided ideas for the development of new treatment strategies [21].

### 3.1. HER-2 Gene Amplification

The propensity for pro-organ metastasis based on molecular subtypes showed increased brain parenchymal colonization and an increased probability of BM in the HER2-overexpressing subtype and triple-negative subtype [22]. Research indicates that BM overexpresses all members of the human EGF receptor (EGFR/HER) family, with HER2 being amplified and overexpressed in approximately 20% of these cases. Furthermore, inhibiting HER may be significant in the therapeutic management of patients with BCBM, even in case of low HER2 gene expression [23]. Therefore, clinical therapy based on HER-2-specific molecules is an important therapeutic direction for the management of BCBM (Figure 2). A range of anti-HER2 therapeutics have been formulated for clinical application, encompassing monoclonal antibodies like trastuzumab and pertuzumab, small-molecule TKIs including lapatinib, neratinib, pyrotinib and tucatinib, and ADCs like T-DM1 and T-DXd. According to clinical guidelines (e.g., NCCN guidelines), trastuzumab and pertuzumab combined with chemotherapy are the first-line treatment options for the treatment of HER2-positive breast cancer patients. However, patients with advanced brain metastasis often have drug resistance to first-line medications, so other treatment options can be tried, such as ADC and TKI drugs. Except for pyrotinib, every drug mentioned have been approved by the FDA for the treatment of HER2-positive breast cancer. And they have shown good efficacy in patients with metastatic HER2-positive breast cancer (Table 1), providing a therapeutic direction for patients with BM, and some of these further retrospective studies, as well as new studies, have shown efficacy of these drugs in BM.

#### 3.1.1. Monoclonal Antibody

##### 3.1.1.1. Trastuzumab

Research have found that breast cancer patients who received trastuzumab exhibited a obviously extended median duration to the progression of BM (15 months compared to 10 months) and a longer median survival time until death (14.9 months versus 4.0 months) than those who didn’t take trastuzumab [39]. In addition, Rostami et al. provided evidence indicating that patients with BM resulting from HER2-positive BC had an extended mean survival following treatment with trastuzumab, reporting a longer mean survival of 17.5 months compared to 11 months for untreated patients.

##### 3.1.1.2. Pertuzumab

Furthermore, the integration of pertuzumab with trastuzumab and paclitaxel has shown enhanced efficacy compared to the conventional treatment regimen of trastuzumab and paclitaxel alone, as evidenced by an increase in median brain metastasis progression-free survival (PFS) from 11.9 months to 15 months. Additionally, the cohort receiving pertuzumab exhibited an extended median overall survival (OS) of 34.4 months, compared to 26.3 months in the control group [40].

#### 3.1.2. TKI

##### 3.1.2.1. Lapatinib

In 2007, the U.S. FDA approved lapatinib combined with capecitabine for HER2-positive advanced or metastatic breast cancer patients. Lapatinib is a first-generation TKI that can reversibly inhibit HER1 and HER2. Its CNS objective response rate (ORR) of lapatinib-based monotherapy is only about 6% [41]. However, LANDSCAPE research has demonstrated that a combination of lapatinib and capecitabine had a CNS ORR of up to 65.9% and a median PFS of 5.5 months in patients with untreated BCBM. The RECIST 1.1 CNS ORR was 57% [42]. These data are the first to indicate the effectiveness of lapatinib in treating BCBM.

##### 3.1.2.2. Neratinib

As early as 2017, neratinib was approved by the US FDA for intensive adjuvant therapy after completion of adjuvant trastuzumab in adult patients with HER2-positive early-stage breast cancer. For patients with brain metastases, clinical trials have also confirmed that neratinib combined with capecitabine is effective for brain metastases in refractory HER2-positive breast cancer [34], and the specific benefit data are shown in Table 1, but about 30% of patients have adverse reactions to diarrhea, and efforts are needed to reduce diarrhea in order to better improve the quality of life of patients.

##### 3.1.2.3. Pyrotinib

Pyrotinib has not been approved by FDA, but clinical trials have also shown good efficacy. In addition to lapatinib, the phase II PERMEATE trial showed the superior therapeutic effect of pyrotinib in conjunction with capecitabine in patients diagnosed with HER2-positive BCBM: the trail included two different patient cohorts: patients exhibiting HER2-positive brain metastases who had not undergone radiotherapy (Cohort A) and those who had experienced disease progression following radiotherapy (Cohort B). The findings indicated that the intracranial ORR of cohorts A and B were 74.6% and 42.1%, and their mPFS time were 11.3 months and 5.6 months, which showed that the combination of pyrotinib and capecitabine had potential anticancer effect in patients who had not undergone radiotherapy [35]. Similarly, another non-randomized study focusing on HER2-positive BCBM patients demonstrated that CNS radiotherapy combined with pyrotinib and capecitabine showed a 1-year PFS rate of 74.9%, a median PFS of 18.0 months, and an ORR of 85% in the CNS [43].

##### 3.1.2.4. Tucatinib

The next-generation TKI drug tucatinib, a TKI specific for targeting HER2, has shown significant efficacy when used in conjunction with trastuzumab and capecitabine for treating patients with BM resulting from HER2+ breast cancer, as evidenced by the findings of the phase III HER2CLIMB clinical trial. The study focused on survivors with BM who had already undergone therapy with trastuzumab, pertuzumab and T-DM1 treatment, starting the patients who had taken tucatinib in conjunction with trastuzumab and capecitabine had an intracranial ORR of 47.3%, a mPFS of 9.9 months, and an mOS of 18.1 months, while the control group who received a placebo in conjunction with trastuzumab and capecitabine had an intracranial ORR of 20.0%, a mPFS of 4.2 months, and an mOS of 12.0 months [44]. This shows that tucatinib is useful for treating patients with BCBM.

#### 3.1.3. ADCs

##### 3.1.3.1. T-DM1

As an increasingly important curative class, ADCs are effective drugs for the treatment of BCBM. Despite the relatively large size of the ADC molecule, it can penetrate the brain and release large quantities of cytotoxic drugs [45]. T-DM1, the earliest ADC sanctioned for BC treatment, consists of trastuzumab and the cytotoxic antimicrotubule compound DM1, interconnected by an irreversible thioether linker [46]. In 2013, the FDA approved it for the treatment of HER2-positive MBC patients who had received trastuzumab and first-line chemotherapy. Then in 2015, a retrospective and exploratory analysis of the EMILIA trial demonstrated an mPFS of 5.9 months (vs. 5.7 months) and mOS of 26.8 months (vs. 12.9 months) for T-DM1 compared to lapatinib in conjunction with capecitabine in BCBM patients previously treated with trastuzumab in combination with paclitaxel [47].

##### 3.1.3.2. T-DXd

T-DXd, a new ADC class, couples a monoclonal antibody targeting HER2 with a topoisomerase I inhibitor via a tetrapeptide shearable linker. In the DESTINY-Breast01 trial, 24 patients with brain metastases who received T-DXd had an average mPFS of 18.1 months, far exceeding the figure for patients receiving the standard treatment [48]. In addition, the DESTINY—Breast03 study again validated the superior efficacy of T-DXd compared to T-DM1 in BCBM patients, with mPFSs of 15.0 and 3.0 months in 114 patients with BCBM in the two groups and ORRs of 67.4% and 20.5%, respectively in 2023 [38].

### 3.2. Aberrant Signaling Pathway

Numerous aberrantly expressed genes and signaling pathways such as the dysregulation of Notch and WNT pathways, has been documented in individuals diagnosed with BCBM [49,50]. WNT and Notch pathways are primitive, conservative evolutionary pathways that determine cell destiny that are highly relevant to many facets of tumor biology, encompassing the sustenance of cancer stem cells (CSCs), angiogenesis, and immunity [51,52]. Therefore, they can lead to tumor formation when they are abnormally activated.

#### 3.2.1. WNT Signaling Pathway

The WNT signaling pathway is overexpressed in individuals diagnosed with BC, particularly in those with BM, and the WNT-selective receptor ROR2 is also overexpressed. The importance of ROR2 in cancer initiation and progression has been demonstrated in vivo in a mouse model of basal-like TP53-deficient BC, and the silence of ROR2 inhibits cancer growth in the mouse brain, suggesting that WNT is related to cancer development and brain metastasis [53].

At the molecular level, the overexpression of the WNT-selective receptor ROR2 upregulates some non-classical WNT ligands, especially WNT11, and ligand-receptor binding regulated various functional proteins, increasing the invasiveness of breast cancer brain metastases [54]. In addition, endothelial WNT signal transduction affects the BBB by modulating the expression of various functional proteins [55,56]. In patients with BCBM, the WNT signal controls the overexpression of the Mfsd2a protein [57,58]. MFSD2A is a sodium-dependent lysophosphatidylcholine cotransporter that regulates the BBB’s function by transporting lipids.

Studies have found that wogonin (WOG), an inhibitor of the WNT pathway [59], reversibly inhibits BBB WNT signaling and reduces Mfsd2a production. Applying nanoparticles for the delivery of WOG and orlistat enhances the intracranial accumulation of orlistat and inhibits tumor progression by releasing orlistat to interfere with tumor lipid metabolism. Rast studies have achieved good efficacy in mice with BCBM, improving the life quality in mice and significantly limiting intracranial tumor progression. The median survival time of mice increased to 24 days (that for nano-orlistat alone was 17 days), and it did not lead to cerebral edema, cognitive deficits, or widespread injury in normal mice [60].

Phase I clinical trials currently utilize the WNT inhibitor LGK974 to treat patients with metastatic colorectal cancer [61]. Porcupine (PORCN) is essential for WNT ligand secretion [62]. LGK974 is a specific inhibitor of porcupine that prevents cancer development by blocking WNT ligand secretion [63]. However, clinical trials investigating WNT inhibitor applications in patients with BCBM are still absent.

#### 3.2.2. NOTCH Signaling Pathway

The Notch pathway is an extremely conservative evolutionary pathway for biologically measuring intercellular interactions and it is capable of interacting with transcriptional regulators in the nucleus to modulate the transcription of target genes that encode transcriptional regulatory proteins and related to some cellular biochemical activities such as differentiation, cell cycle progression [64].

Earlier studies have shown that BCCs with significant brain metastasis tendencies increase the activity of the Notch pathway through Notch1 and Jagged-2 (JAG2). In addition, the Jagged classical Notch ligand 1 (JAG1) has been proven to be overexpressed in metastatic breast cancer tissues, especially brain metastasis. JAG1 also induced Notch signaling in neighboring cells, and the activation of the JAG1-Notch pathway promoted the EMT of BCCs via the upregulation of transcriptional repressors, such as Slug. In addition, in a BBB model, the inhibition of the JAG1-Notch pathway by knocking down JAG1 significantly attenuated the ability of BCCs to penetrate the BBB, inhibiting their proliferation, migration, and invasion, thus suggesting that the JAG1-Notch pathway promotes the development of brain metastasis [65].

Notch pathway inhibitors developed based on this principle can be used as therapeutic methods for patients with BCBM. Researchers have demonstrated that Compound E, a γ-secretase inhibitor with BBB permeability, as a Notch inhibitor, can significantly inhibit brain metastasis in an in vitro mouse model; dominant-negative MAML (dnMAML), a Notch-inhibitory peptide, is also a potent Notch inhibitor, and it significantly attenuates the growth of BCCs metastasizing in the brain [66]. However, the regulatory mechanism of the Notch signaling pathway is complex, and the best way to accurately regulate it to achieve precise therapeutic effects is still unknown, so its applications in clinical practice still needs to be extensive studied.

#### 3.2.3. CDK4/6 Pathway

The cell cycle is the most significant method through which mammalian cells control proliferation. The G1 phase of the cell cycle is a critical phase of cell division that marks the beginning of the preparatory phases of growth and DNA replication following the completion of the last division, and this restriction point is key for regulating cell proliferation. This restriction point is regulated by the retinoblastoma pathway (CDK4/CDK6-cyclin D1-Rb-p16 / ink4a) [67]. The cell cycle protein D1 (cyclin D1) is overexpressed in most human BCCs, and cyclin D1 stimulates BCC proliferation by activating the cell cycle protein-dependent kinase 4/6 (CDK4/6) [68], whereas CDK4/6 inhibition induces cell cycle arrest in G1 cells, thereby inhibiting BCC proliferation [69]. CDK4/6 inhibitors developed based on this principle have made significant contributions to BC treatment. Currently, clinical studies are assessing the effects of CDK4/6 inhibitors in BCBM patients.

One CDK4/6 inhibitor, abemaciclib, was approved by the U.S. Food and Drug Administration (FDA) in 2017 for use in conjunction with endocrine therapy for breast cancer treatment. In general, for the systemic treatment of BM, drug delivery into the CNS is a major obstacle. However, preclinical studies have found that abemaciclib can traverse the BBB in mice and reach the CNS at an effective concentration, justifying its application in patients diagnosed with BCBM [67].

Past clinical trials have shown an intracranial ORR of 5.2%, an intracranial clinical benefit rate (CRB) of 25%, a median intracranial PFS of 4.9 months, and a median OS of 12.5 months in a cohort of 58 HR+/HER2-BCBM patients. Additionally, in this experiment, the average concentration of the drug in patient’s cerebrospinal fluid (CSF) exceeded the half-maximal inhibitory concentrations (IC50) of CDK4 and CDK6 21 times and 4.3 times, respectively. Thus, these data proved that abeciclib and its active metabolites reached pharmacologically relevant concentrations in the brain and CSF of patients. These results further support the use of abemaciclib for treating patients [70].

#### 3.2.4. Trophoblast Cell Surface Antigen-2 (Trop-2)

Trop-2 is a differentially expressed cell surface glycoprotein in epithelial tumors [71,72]. Trop-2 expression is elevated in breast cancer cells, stimulating cancer cell growth and enhancing tumor aggressiveness [73,74]. The Kaplan–Meier survival curve showed that patients with breast cancer with high Trop-2 expression had significantly shorter survival times [74]. An analysis of the breast cancer genome revealed that Trop-2 was significantly overexpressed in triple-negative breast cancer (TNBC), suggesting that Trop-2 may be a potential target for TNBC [75]. However, normal brain tissue does not express Trop-2 [76], so Trop-2 may be a potential therapeutic target for BCBM.

An antibody-drug conjugate, sacituzumab govitecan (SG), has been developed for Trop-2 targets, consisting of an antibody targeting Trop-2 conjugated to SN-38 (topoisomerase I inhibitor) through a hydrolyzable linker. In a phase III clinical trial, for patients with metastatic TNBC, mPFS was 5.6 months versus 1.7 months and mOS was 12.1 versus 6.7 months for patients with SG versus chemotherapy [77], proving the excellent efficacy of SG.

Recently, a phase 0 window-of-opportunity trial of SG in BCBM patients showed that SG could achieve intratumoral concentrations of sufficient SN-38 to provide therapeutic benefit. In addition, a mouse model of the intracranial inoculation of triple-negative breast cancer cells verified that SG inhibited intracranial tumor growth, and the animal model showed that the control animal group had rapid tumor growth, with all animals dying at 45 days. However, the group of animals treated with SG had a reduced tumor burden, and all animals remained alive at 60 days. Moreover, the investigation encompassed 13 BCBM patients administered an intravenous dose of SG (10 mg/kg) the day prior to surgical intervention, followed by treatment in 21-day cycles, with doses received on days 1 and 8 of each cycle, commencing on postoperative day 1. An analysis of postoperative tissue samples revealed a median total concentration of SN-38 in BCBM of 249.8 ng/g, indicating that SG significantly penetrates intracranial tumors and exhibits promising efficacy within the CNS. The secondary endpoint of the study indicated an OS of 35.2 months, with PFS recorded at 8 months with an ORR of 38%. Additionally, an exploratory endpoint revealed that Trop-2 expression was present in 100% of the BCBM cases examined. The study provides strong support and evidence for treating BCBM with Trop-2 ADC and has important clinical significance [78].

#### 3.2.5. Programmed Cell Death Protein-1 (PD-1)/Programmed Cell Death Ligand 1 (PD-L1)

PD-1 is a type I transmembrane protein composed of 288 amino acid residues, belonging to the CD28 family. It is widely expressed on various immune cells, including activated T cells, B cells, NK cells, and dendritic cells (DCs) [79]. Structurally, PD-1 comprises an extracellular immunoglobulin variable (IgV) domain, a hydrophobic transmembrane domain, and a cytoplasmic tail domain containing immunoreceptor tyrosine-based inhibitory motifs (ITIMs) and immunoreceptor tyrosine-based switch motifs (ITSMs) [80].

PD-L1, the ligand for PD-1, consists of 290 amino acid residues and is primarily composed of a short cytoplasmic tail region, a transmembrane region, and extracellular IgV and IgC-like domains. PD-L1 is expressed on tumor cells and antigen-presenting cells (APCs), including dendritic cells and macrophages. Additionally, PD-L1 expression can be induced on vascular endothelial cells in response to interferon-gamma (IFN-γ) stimulation [81]. Upon binding of PD-1 on T cells to PD-L1 on tumor cells or APCs, the ITIMs and ITSMs of PD-1 are phosphorylated, leading to the recruitment and activation of the protein tyrosine phosphatase SHP-2 (Src homology region 2 domain-containing phosphatase 2). This process inhibits T-cell activation, promotes T-cell apoptosis, reduces cytokine production, and induces antigenic tolerance, thereby facilitating tumor immune evasion [82].

BC has traditionally been considered a non-immunogenic tumor. However, recent studies have revealed that PD-L1 expression and tumor-infiltrating lymphocytes (TILs) are significantly higher in TNBC patients compared to other breast cancer subtypes, suggesting that TNBC exhibits strong immunogenicity and may be amenable to anti-PD-1/PD-L1 therapy [83,84]. Currently, pembrolizumab (a PD-1 inhibitor) and atezolizumab (a PD-L1 inhibitor) have been approved by the FDA for the treatment of PD-L1-positive, unresectable, locally advanced, or metastatic TNBC.

Clinical trials such as KEYNOTE-355, KEYNOTE-522, and KEYNOTE-173 have demonstrated promising efficacy and manageable adverse effects of pembrolizumab in the neoadjuvant treatment of early-stage TNBC at high risk of recurrence, as well as in TNBC patients resistant to multiline chemotherapy regimens [85,86,87]. However, these trials did not include patients with BM. Emerging preclinical and clinical studies support the use of anti-PD-1/PD-L1 therapy in patients with BM [88,89]. A preclinical study demonstrated that sequential in situ immunomodulation treatment combined with anti-PD-L1 therapy exhibited synergistic antitumor efficacy and improved survival in a mouse model of BCBM [88]. A phase II clinical trial involving 35 patients with BCBM (encompassing all four breast cancer subtypes) reported that 37% of patients achieved intracranial benefit from pembrolizumab treatment, with some experiencing mild adverse effects such as nausea, vomiting, and headache [89]. These findings suggest that anti-PD-1/PD-L1 therapy holds promise for patients with BM, although further trials are needed to validate these results.

The exploration of neoadjuvant therapy for early-stage BC in combination with immune checkpoint inhibitors remains limited but ongoing. For instance, an experimental approach combining the PD-L1 inhibitor atezolizumab with trastuzumab and pertuzumab was attempted in patients with HER2-positive BCBM. Unfortunately, no significant CNS benefits were observed in these patients [90]. Therefore, the potential benefits of PD-1/PD-L1 inhibitors for patients with BM require further investigation through additional studies.

#### 3.2.6. In Summary

The treatment landscape for breast cancer brain metastases (BCBM) is evolving rapidly, with significant advancements in targeted therapies and a deeper understanding of molecular mechanisms. The following points summarize the current status and future directions for therapeutic strategies. Nowadays, anti-HER2 treatments have demonstrated substantial clinical benefits in patients with HER2-positive BCBM. Drugs such as trastuzumab, pertuzumab, lapatinib, neratinib, tucatinib, T-DM1, and T-DXd have shown significant efficacy in prolonging progression-free survival (PFS) and overall survival (OS) in clinical trials. The use of anti-HER2 therapies is well-established and recommended for patients with HER2-positive BCBM. Additionally, WNT and Notch signaling pathways play a crucial role in tumor biology and metastasis. However, their mechanisms are highly complex and diverse. Current research on WNT and Notch inhibitors is primarily at the preclinical stage. While these pathways show promise as therapeutic targets, further studies are needed to elucidate their roles and develop effective inhibitors for clinical use. In addition, the CDK4/6 inhibitor abemaciclib has shown promising results in HR+/HER2- BCBM patients. With its ability to cross the blood-brain barrier (BBB) and achieve effective concentrations in the brain, abemaciclib represents a new direction for the treatment of BCBM. Ongoing clinical trials are further exploring its potential in this patient population. Sacituzumab govitecan (SG), trop-2-targeted ADC, has demonstrated significant efficacy in preclinical models and clinical trials with small sample sizes. The promising results in both mice and patients suggest that SG could be a valuable treatment option for BCBM. Larger-scale clinical trials are warranted to confirm its efficacy and safety in a broader patient population. Anti-PD-1/PD-L1 therapies have achieved remarkable success in non-brain metastatic triple-negative breast cancer (TNBC) patients. However, data on their efficacy in patients with brain metastases are limited. Preclinical studies and early clinical trials have shown some potential benefits, but more research is needed to determine their role in the treatment of BCBM.

In conclusion, while significant progress has been made in the treatment of BCBM, there are still many challenges to overcome. Future research should focus on developing more effective targeted therapies.

## 4. Therapeutic Strategies Based on BBB Drug Delivery Pathways

The BBB regulates the homeostasis of CNS by establishing a meticulously controlled neurovascular unit (NVU) that preserves proper brain function, and it has an intact endothelium that can act as a defense barrier against tumor-cell extravasation; however, these properties also hinder drug delivery to the brain. During the invasion and infiltration of BCCs into the brain, the integrity of the BBB is disrupted and permeability is increased, but the TJ undergoes rapid remodeling during TEM and does not significantly damage the BBB [91]. In addition, the BBB has characteristics such as active efflux molecules [15]. Therefore, achieving drug delivery across the BBB is a significant issue when treating brain metastases. We need to discuss emerging strategies for enhancing medication transport across the BBB for the improved treatment of patients with BCBM.

### 4.1. Molecular Pathway

#### 4.1.1. LRP-1

Receptor-mediated transcytosis (RMT) is now used for the noninvasive delivery of drugs across the BBB. Proteins expressed by ECs on the BBB can be used for receptor-mediated BBB penetration of the cytosol, e.g., the transferrin receptor, insulin receptor, insulin-like growth factor 1 receptor, and LDL receptor [92]. In this case, binding the targeted ligand (e.g., a monoclonal antibody) to its receptor triggers endocytosis, followed by intracellular vesicular translocation and eventual release into the brain, and using this translocation mechanism, the drug can be molecularly ligated to this ligand to achieve translocation across the BBB via receptor-mediated endocytosis, e.g., by using endogenous expression of the BBB receptor, the low-density lipoprotein receptor-related protein-1 (LRP-1) [93], a recent RNA sequencing study found that LRP-1 mRNA was expressed at higher levels than the insulin receptor and TfR mRNA in isolated human cerebral vessels [94].

ANG1005 is a peptide–chemotherapy coupled drug targeting LRP1, consisting of three paclitaxel molecules covalently attached to Angiopep-2 (a peptide engineered to exploit the LRP1 transport mechanism), which increases the spread of drugs to brain metastases through RMT (Figure 1). Phase II trials have found that ANG-1005 shows activity against breast cancer and brain tumors. This phase II clinical trial enrolled 72 female BCBM patients, of whom 57% (41) were HER2-negative and 19 (26%) were TNBC patients. The findings showed that ANG 1005 demonstrated significant CNS efficacy, achieving an average intracranial ORR of 15% and an intracranial CBR of 68% for every patient. The median intracranial PFS and OS were recorded at 2.8 months and 7.8 months, respectively [95]. Currently, a phase III trial on ANG1005 is underway [96].

#### 4.1.2. Overcoming the Efflux Pumps

BCBM cancer cells express greater amounts of breast cancer resistance protein (BCRP) and P-glycoprotein (P-gp) than cancer cells derived from the primary tumor [97]. P-gp and BCRP are efflux pump transport proteins located at the BBB and restrict the uptake of substrate drugs into the brain, which can exclude the drug from the cytoplasm [98]. They do this by actively pumping out the drug so that drug concentrations in the brain reduce and drug levels fall well below the threshold necessary for therapeutic efficacy, leading to drug resistance. The current study suggests that P-gp excludes drugs from the cell via a twist–squeeze mechanism. P-gp is embedded in the phospholipid bilayers of biological membranes and has two homologous transmembrane structural domains (TMDs) that bind substrates and two intracytoplasmic nucleotide-binding structural domains (NBDs) that bind ATP. The tilt–shift of the two TMDs gives P-gp two conformations: the inward-facing type provides binding sites for intracellular drugs, while the outward-facing type promotes the efflux of bound drugs. When binding only ATP, P-gp can more stably maintain the inward conformation, and after binding the substrate again, the whole protein twists and moves, and the inward type is converted to the outward type, which directly squeezes the substrate to the outside of the cell [99]. Currently, many drugs exist that transport substrates for transfer proteins, such as the tyrosine kinase inhibitors (TKIs) lapatinib, erlotinib [100], and temozolomide, in which transporter proteins reduce the brain penetration of temozolomide and thus its anti-tumor efficacy [98]. Additionally, P-gp can transport paclitaxel, which contributes to the development of resistance to this drug [101].

Therefore, to combat intracerebral drug decreases mediated by efflux transporters, reduce drug resistance, and achieve intracranial effective concentration, preclinical investigations have concentrated on using transporter inhibitors to develop drugs. For example, one study has found that encequidar (HM30181A), a P-gp inhibitor, can enhance the uptake of paclitaxel and the anti-tumor effect of paclitaxel in an orthotopic brain tumor model [102]. A clinical trial (NCT02594371) demonstrated that patients with MBC receiving oral paclitaxel in combination with encequidar exhibited superior PFS and OS compared to those treated with intravenous paclitaxel. Specifically, the oral paclitaxel and encequidar cohort achieved a PFS of 8.4 months, while the intravenous paclitaxel group had a PFS of 7.4 months. Furthermore, the OS for the oral paclitaxel and encequidar group was recorded at 22.7 months, in contrast to 16.5 months for the intravenous paclitaxel group [103].

Utidelone (UTD-1) inhibits multidrug resistance through multiple pathways, including the inhibition of P⁃gp-mediated mutations in the paclitaxel-binding region and the overexpression of transporter proteins [104]. Currently, the FDA has approved UTD-1 for use in patients with BCBM. At the 2024 American Society of Clinical Oncology (ASCO) meeting, an experiment (NCT05357417) investigating the combination of UTD-1 and bevacizumab for treating brain metastases from TNBC achieved a CNS ORR of 55% among participants. Additionally, the study reported a median PFS of 8.4 months, confirming that UTD-1 had good anti-tumor activity against brain metastases [105]. Another trial indicated that patients with MBC who exhibited resistance to anthracycline and paclitaxel experienced improved PFS, OS, and ORR when treated with UTD-1 plus capecitabine compared to those receiving capecitabine monotherapy. Specifically, the OS was 19.8 months for the combination therapy group, in contrast to 16 months for the group receiving capecitabine alone. The assessment of the PFS and ORR showed that the combination group had a median PFS of 8.6 months and an ORR of 49.8%, while the monotherapy group had a median PFS of 4.1 months and an ORR of 26.7%, which demonstrated the superiority of UTD-1 used in combination with capecitabine treatment over capecitabine monotherapy [106].

#### 4.1.3. VEGF

The expression of VEGF was elevated in BCCs isolated from brain metastases in a preclinical experiment [107]. VEGF induces neovascularization by binding to VEGFR2 and activating the STAT3 and PI3K pathways, thereby enhancing vascular permeability, and downregulates vascular endothelial integrity to enhance brain metastasis in tumor cells [108].

Bevacizumab was the first targeted anti-angiogenic agent developed, and an E2100 study in patients with advanced MBC demonstrated that patients in the paclitaxel–bevacizumab group had nearly 1-fold longer PFS than patients in the paclitaxel-alone group (mPFS 11.8 months vs. 5.9 months) [109]. The AVADO study demonstrated that PFS was much longer in patients with MBC who received bevacizumab in conjunction with docetaxel compared to those in the control group [110]. Similarly, another study (RIBBON-1) focusing on patients with MBC showed that patients receiving bevacizumab in conjunction with either capecitabine or a regimen of paclitaxel/anthracycline experienced a statistically obvious improvement in PFS compared to those administered chemotherapy alone. Specifically, the median PFSs were reported as 8.6 months versus 5.7 months and 9.2 months versus 8.0 months, respectively [111]. Nevertheless, as the study progressed, an analysis of the efficacy and safety data derived from these three clinical studies revealed that bevacizumab was only effective at prolonging PFS among patients; it did not demonstrate an improvement in OS. Additionally, the proportion of serious adverse reactions such as intracerebral hemorrhage increased [109,110,111]. Consequently, the authorization of bevacizumab use in conjunction with chemotherapy as a first-line therapy for advanced BC was rescinded.

Nonetheless, bevacizumab, an antibody targeting VEGF, as a humanized IgG1 monoclonal antibody that binds to VEGF with high affinity, is of clinical value in patients with BCBM. Bevacizumab showed favorable efficacy in a phase II trial in conjunction with carboplatin for treating patients with BCBM. The CNS ORR was 63%, median PFS was 5.62 months, and median OS was 14.10 months, with efficacy results exceeding historical data for platinum agents alone [112]. Another clinical trial with 112 patients examined whether adding bevacizumab, etoposide, and cisplatin (BEEP) prior to whole-brain radiotherapy (WBRT) improved brain-specific PFS. It was found that median brain-specific PFS was 8.1 and 6.5 months in the WBRT and BEEP group and the WBRT-alone group, respectively. However, there was no notable difference observed in brain-specific ORR at the two-month mark (BEEP and WBRT vs. WBRT, 41.9% vs. 52.6%). Meanwhile, the rate of brain-specific PFS was 48.7 versus 26.3 percent at 8 months. These findings suggest that BEEP induction chemotherapy prior to WBRT may inhibit brain metastasis progression compared to WBRT alone [113]. These data reveal the value of bevacizumab, an antibody targeting VEGF, in treating BCBM.

Targeting the VEGF-VEGFR system, PTK787, an orally effective selective VEGFR inhibitor, obviously limited angiogenesis and brain metastasis growth and increased apoptosis in tumor cells in preclinical models (Figure 1).

Nevertheless, there was no notable enhancement in survival rates among the treated mice, possibly associated with cancer cells being able to express other cytokines, such as IL-8 [114]. IL-8 acts as a mediator of angiogenesis and induces endothelial cell proliferation, which induces angiogenesis even when VEGF receptor signaling is impaired, thereby promoting the progression of brain metastases [107]. After the infiltration of cancer cells into the brain, brain metastatic cells secreted VEGF and IL-8 in significantly higher contents than parental cells [22]. In contrast, AM9928, a potent covalent monoacylglycerol lipase (MAGL) inhibitor, was found to inhibit the secretion of IL-8 and VEGF in TNBC cells, thereby reducing TNBC colonization in the brain [115]. These outcomes imply that AM9928 can be used as a new drug for patients with TNBC brain metastases.

### 4.2. Physicochemical Pathway

Currently, multiple modalities of electromagnetic radiation and ultrasound are utilized to physically disrupt the BBB. For example, laser-induced interstitial thermotherapy (LITT) employs an implanted laser to achieve localized opening of the BBB, promoting the entry of antineoplastic agents into the rat brain [116]. Another approach is Focused Ultrasound (FUS), found in preclinical studies to allow easier drug passage through the BBB. FUS is associated with a microbubble system, where ultrasound penetrates deeper into soft tissues. The mechanical interactions between ultrasound, microbubbles, and the vascular system lead to the intermittent depolymerization of TJ proteins on the ECs of the BBB [117]. This process provides a transient opportunity to administer therapeutic medicines. The FUS-mediated remodeling of the BBB has been applied for the management of brain cancers located in various regions of the CNS. This technique enhances the permeability of a variety of antineoplastic agents such as temozolomide and paclitaxel, aiding their translocation across the BBB [118]. Preclinical experiments are now available to demonstrate that FUS and microbubbles in combination with weekly trastuzumab therapy enhance BBB permeability, improving efficacy in mice inoculated with HER2-positive mammary tumors in the brain, notably decreasing the average intracerebral tumor volume, as well as a substantially enhancing the survival duration of the mice [119]. Moreover, chemical irritation may induce endothelial contraction, leading to the instantaneous destruction of the BBB reversibly, as exemplified by a study on hypertonic mannitol solution [117].

### 4.3. Application of Nanoparticles

Delivering generic drugs to brain tissue remains a challenge due to the BBB. For patients with BCBM, drug delivery system (DDS) must overcome this obstacle. Nanoparticles, as a class of ultrafine materials, exhibit unique physicochemical properties due to their extremely small sizes: as carriers of therapeutic drugs, they are more easily solubilized and encapsulated [119]. Therefore, nanoparticle-based DDS could achieve the efficient penetration of the BBB [120]. This means that patients with BCBM have a promising future in nanotherapeutics.

Many nanoparticles are being investigated for treating BCBM, including carbon-based, lipid-based, polymer-based, and metal-based nanostructures [121]. Compared with free medications, nanostructures can also selectively deliver medications to cancerous sites, stimulate more apoptosis in breast tumors, and significantly reduce the systemic toxicity of drugs. Research has demonstrated that nanostructures can facilitate the delivery of a diverse array of drugs; for example, metal nanoparticles and carbon nanoparticles can transport cisplatin to reduce its toxicity and enhance therapeutic efficacy [122]. Liposomal nanocarriers can be used to deliver doxorubicin; albumin nanoparticles in polymeric nanoparticles that can be used to deliver paclitaxel [121]. In addition, lipid nanoparticles can carry docetaxel. An animal study found that BCCs can rapidly absorb the docetaxel-loaded lipid nanoparticles. Moreover, docetaxel has higher brain bioavailability in this form, significantly inhibiting the development of BM and prolonging survival in a mouse model [123]. A preclinical animal study demonstrated that mice with BM receiving liposomal irinotecan achieved a superior survival rate compared to free irinotecan alone [124]. In addition to preclinical trials, a phase I clinical trial investigated the effects of liposomal irinotecan. This research found that in seven of these BCBM patients, tumor shrinkage ranged from 7% to 55%, the ORR in BCBM patients was 30%, and 50% of the patients achieved a significant clinical benefit [125].

In addition to delivering drugs, nanotechnology can be used for immunotherapy. The tumor microenvironment is a dynamic system that includes tumor cells, the extracellular matrix, and mesenchymal tissues. It is a key factor affecting tumor metastasis. Moreover, in the brain tumor microenvironment of BCBM patient, tumor-associated macrophage (TAM) represents the predominant cell type within the metastatic microenvironment [126]. TAM has both pro-inflammatory M1-like and anti-inflammatory M2-like phenotypes [127]. However, the majority of TAM belong to M2-like phenotypes, which can facilitate angiogenesis and reshape the extracellular matrix, thereby promoting tumor progression [128]. Therefore, when treating patients with BCBM, if TAM can be reprogrammed from the M2 to the M1 phenotype, it will enhance the tumor immune response.

Many experiments have studied the nanostructure of immunotherapy for BCBM. In detail, some studies have used nanoparticle-encapsulated drugs to target receptors on the TAM surface to inhibit TAM or to polarize M2-like TAM to M1-like TAM [129]. For example, dextran-coated magnetic nanoparticles (Ferumoxytol) can significantly inhibit breast cancer progression and metastasis in mice by generating reactive oxidative species (ROS) via the Fenton reaction that mediates the repolarization of TAM to the anti-tumor M1-like form [130]. In addition, copper sulfide nanoparticles (CuS NPs) increase intracellular ROS levels and facilitate the polarization of TAM to a M1-like phenotype in mice [131]. A study showed that binding SIRPα and CD47 on macrophages prevented macrophages from engulfing tumor cells [132]. The membrane-encapsulated magnetic nanostructure can obstruct the interaction between SIRP α and CD47, thereby maintaining the activity of macrophages to engulf breast cancer cells. Meanwhile, the magnetic core facilitates the repolarization of M2 macrophages to M1 macrophages. These combined effects contribute to the enhancement of immune responses and subsequently improve survival rates and prognostic outcomes for patients with breast cancer [133].

In addition, nano can induce photothermal therapy (PTT) to promote tumor cell death; gold nanoparticles of different sizes and dimensions can use different wavelengths of light to perceptually heat tumors by altering their photothermal and photoacoustic behaviors, killing breast cancer cells by heating them without overheating normal cells [134].

### 4.4. Direct Infiltration

Temozolomide (TMZ), classified as a second-generation oral alkylating agent, can pass through the BBB readily. It mainly attacks the DNA of tumor cells, causing DNA alkylation damage and then forming DNA cross-links, leading to the death of cancer cells. TMZ is rapidly absorbed after oral administration and then crosses the BBB into the CSF, reaching an effective drug concentration [135].

However, a phase II trial [136] demonstrated that TMZ, administered orally to treat patients with brain metastases, resulted in transient stabilization in only three patients (durations of 2.7 to 5.6 months), with disease progression recorded in the remaining 15 patients. For five patients with brain metastases, temozolomide did not show any significant improvement in brain lesions. Another phase II clinical trial investigating WBRT combined with temozolomide in patients with BCBM showed an ORR of 36% versus 30% in the WBRT group versus WBRT and TMZ group at 6 weeks, a median PFS of 7.4 months versus 6.9 months, and a median OS of 11.1 months versus 9.4 months [19]. These experiments demonstrate that TMZ has limited potential for treating BCBM.

## 5. Discussion

This article details therapeutic strategies for BCBM, specifically drug therapy based on molecular mechanisms and delivery pathways across the BBB.

The molecular mechanisms underlying brain metastasis are intricate and encompass aberrant expression of numerous genes and signaling pathways. For instance, the HER2 gene is amplified, and the use of anti-HER2 drugs has been associated with enhanced patient survival in clinical settings. However, trastuzumab and pertuzumab are common with side effects such as cardiac dysfunction, rash, and fever. A trial of safety and tolerability in patients with HER2-positive breast cancer showed a higher incidence of diarrhoea and a higher incidence of neutropenia in the pertuzumab group than in the placebo group [137]. TKI drugs (such as lapatinib, pyrotinib, tucatinib) may also cause diarrhea, abnormal liver function, rash, etc. The combination of a TKI with trastuzumab is more likely to cause diarrhea than trastuzumab alone [138]. CDK4/6 inhibitors (such as abeciclib) may also cause neutropenia, leukopenia, anemia, etc. Long-term use may also cause adverse reactions such as fatigue, nausea, and vomiting [139].

ADC drug T-DXd may cause nausea, vomiting, fatigue, hair loss, etc. However, overall T-DM1 and T-DXd were manageable and safe for longer treatment durations [38]. Common treatment-related adverse events with sacituzumab govitecan (SG) include neutropenia, leukopenia, and anemia. But it is well tolerated in patients with metastatic triple-negative breast cancer who have received significant prior therapy [140]. In addition, PD-1/PD-L1 inhibitors may cause immune-related adverse events, such as immune pneumonia, immune hepatitis, and immune thyroid dysfunction [141].

Due to the complex mechanism of the signaling pathways and the involvement of multiple factors such as hypoxia in their activation, the precise role of activated Notch signaling in the brain-specific metastasis of tumor cells remains unclear [49]. Preclinical trials related to this have shown the potential to inhibit tumor growth, but its role still lacks clinical trial data to confirm. In addition, brain metastases from breast cancer may acquire driver mutations after metastasis to the brain and carry mutations not detected in samples from the primary site [142,143]. However, due to the small sample size, it is difficult to extract general evolutionary patterns between primary and metastatic foci, and biopsies of brain metastases in particular are more difficult. Currently, CSF interacting directly or indirectly with brain tumors may be a new direction for “liquid biopsies” based on brain tumor genetics [144].

The BBB restricts the entry of drugs into the CNS, such as trastuzumab and pertuzumab, due to their large molecular weight, making it difficult for them to cross the BBB. As a result, the concentration of these drugs in the CNS is significantly lower than in other parts of the body. Even if a drug successfully penetrates the BBB, tumor cells may develop resistance through multiple mechanisms, such as P-gp and LRP-1-mediated drug efflux. Therefore, our research is directed towards the development of enhanced drug delivery strategies that take into account the characteristics of this barrier. Examples include the utilization of RMT and the deployment of transporter protein inhibitors to augment the intracerebral concentration of the drug, thereby enhancing its therapeutic efficacy. Additionally, the article delineates various techniques for physically disrupting the BBB, as well as the utilization of nanocarriers (e.g., liposomes and gold nanoparticles) for the targeted delivery of drugs, ensuring efficient delivery to the tumor site while minimizing systemic toxicity. Despite diligent attempts to utilize a variety of nanocarrier, the anti-tumor efficacy of these carriers and their cumulative effect in the targeted tissues are still limited. To solve these challenges, various surface modifications (e.g., folic acid, histidine, transferrin, etc.) and stimulations (e.g., P and, enzymes, photothermal, etc.) have been applied to nanocarriers, but it is still a problem to accurately identify clinically applicable technologies from the wide variety of preclinical trials [133].

In addition, topical therapy is indispensable as an important complement to drug therapy. However, surgery is usually mainly applied to a single lesion and has limited effect on multiple brain metastases. WBRT may lead to irreversible cognitive impairment, with an increased risk of cognitive loss and a significant reduction in patients’ quality of life with prolonged treatment. Although SRS has relatively few side effects, it is only suitable for brain metastases with 1~3 metastases and lesions ≤ 3~4 cm in diameter, and it is still difficult to apply it multiple times [145].

## 6. Future Direction

With the deepening of research on the molecular mechanisms of brain metastasis, more targeted therapies targeting key signaling pathways and genes are being developed. There is still considerable scope for the development of targeted drugs based on HER2, WNT, Notch and other signaling pathways. In addition, three phosphoinositide 3-kinase (PI3K) inhibitors (alpelisib, buparlisib, and dactolisib) in combination with anti-estrogen agents are presently undergoing assessment for their efficacy in the treatment of hormone-positive BCBM. There are also various chemotherapy drugs and targeted drugs in combination with good results. For example, LANDSCAPE study showed that the combination of lapatinib and capecitabine in the first-line treatment of HER2+ BCBM demonstrated a CNS ORR of 65.9% in patients [42]. In addition, P-gp inhibitors can be used in combination with antineoplastic drugs to improve the activity of intracranial antineoplastic drugs and slow down the emergence of drug resistance. Paclitaxel and encequidar has shown promising effect in patients with MBC [103]. And the transporter inhibitor UTD-1 combined with capecitabine and bevacizumab showed anti-tumor activity and controllable toxicity in patients with brain metastases [105,106]. Therefore, the combination of UTD-1 and targeted drugs may be another new idea to overcome drug resistance and enhance drug efficacy, and it is worth conducting randomized controlled trials.

Improving drug delivery systems: Technologies that break through the BBB will be key to the therapy of brain metastases. Using new drug delivery systems like nanotechnology, combined with physical interventions (such as ultrasound and laser), can significantly enhance the penetration and intracranial efficacy of drugs.

The in-depth study of the molecular typing of breast cancer and the mechanism of brain metastasis will facilitate the development of personalized treatment options. The future treatment of patients will be tailored according to their molecular characteristics, with the most appropriate targeted drugs and treatments selected. Although the molecular mechanism of BCBM is currently understood to some extent, further clinical trials are required to verify the efficacy and safety of new drugs and treatment strategies. In the future, the combination of further clinical trials and real-world data will facilitate the determination of the optimal treatment and strategy. In general, the treatment is developing in a more precise and personalized direction, and with the advancement of drug delivery technology and the in-depth understanding of molecular mechanisms, the survival rate of breast cancer brain metastases patients is expected to be further improved in the future.

## Figures and Tables

**Figure 1 pharmaceuticals-18-00262-f001:**
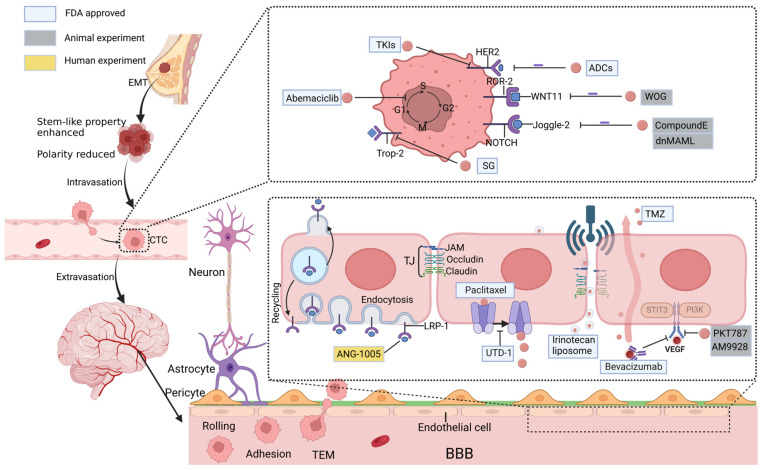
Breast cancer cells metastasize to the brain. Breast cancer cells undergo epithelial-mesenchymal transition (EMT) to become invasive. The invasive cancer cells then enter the circulatory system and become circulating tumor cells (CTCs) with the help of macrophages and endothelial cells (ECs). CTCs spread through the bloodstream to the brain, where they migrate transendothelially (TEM) by rolling, adhering to, and finally crossing the blood-brain barrier (BBB), a specialized structure made up of endothelial cells (ECs), pericytes, basement membranes, and astrocytes. Tight junctions (TJ) between endothelial cells limit paracellular transport to the brain. However, CTC induces disruption of the TJ, enhances BBB permeability, and facilitates CTC extravasation to the brain. Pericytes can also maintain BBB function by regulating TJ formation in the EC and by secreting components of the endothelial basement membrane.

**Figure 2 pharmaceuticals-18-00262-f002:**
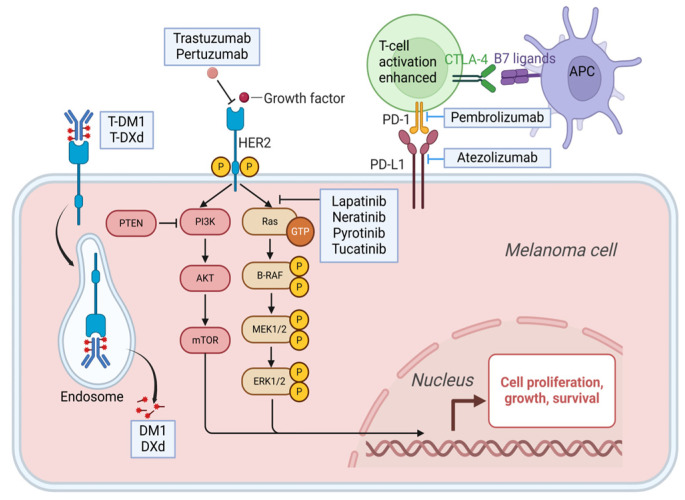
Mechanism of action of anti-HER2 drugs. The HER family includes HER1, HER2, HER3 and HER4, among which HER2 overexpression is the most common in breast cancer patients. HER is a typical cell membrane receptor tyrosine kinase, and due to HER2 overexpression, heterodimerization of this receptor with other members of the EGFR family leads to autophosphorylation of tyrosine residues within the cytoplasmic domain of heterodimeric and initiation of Akt, MAPK, and various other signaling pathways leading to cell proliferation and tumorigenesis [24]. As early as the 90s of the last century, the anti-HER2 monoclonal antibody trastuzumab has achieved good clinical efficacy [25], and then the new monoclonal antibody Pertuzumab inhibits the MAPK and PI3K signaling pathways by inhibiting HER2 dimerization and inhibiting the growth and development of tumor cells [26]. With the development of antibody-drug conjugates, HER2 antibody conjugates T-DM1 and T-DXd have also achieved good efficacy, both of which enhance the killing effect on tumor cells by attaching small molecule toxin drugs to HER2 monoclonal antibodies and then transporting the toxin drug into the cell [27,28]. Lapatinib, pyrotinib, and tucatinib, as TKIs, can also inhibit tumor development by inhibiting the MAPK signaling pathway [29,30].

**Table 1 pharmaceuticals-18-00262-t001:** Major clinical trials of HER2-targeting drugs in patients with metastatic breast cancer. “---” indicates that this information is not mentioned in the experiment.

Drug Classification	Drug Name	Research Title	Study Design	Phase	ORR	OS	PFS
Monoclonal antibodies	Pertuzumab	PATRICIA [31]	Pertuzumab plus high-dose trastuzumab	Phase II	Clinical benefit rates at 4 months and 6 months were 68% and 51%	---	---
		CLEOPATRA [32]	Pertuzumab, trastuzumab, and docetaxel vs. placebo, trastuzumab, and docetaxel	Phase III	---	57.1 months(Pertuzumab) vs. 40.8 months(placebo)	53.8 months(pertuzumab) vs. 46.6 months(placebo)
TKI	Lapatinib	ELTOP [33]	Lapatinib plus capecitabine vs. trastuzumab plus capecitabine	Phase II	41% (lapatinib plus capecitabine) vs. 40% (trastuzumab plus capecitabine)	--- vs. 6.1 months	7.1 months (lapatinib plus capecitabine) vs. 6.1 months (trastuzumab plus capecitabine)
	Neratinib	TBCRC 022 [34]	Lapatinib-naïve vs. lapatinib-treated added to neratinib, capecitabine	Phase II	49% (lapatinib-naïve) vs. 33% (lapatinib-treated)	13.3 months (lapatinib-naïve) vs. 15.1 months (lapatinib-treated)	5.5 months (lapatinib-naïve) vs. 3.1 months (lapatinib-treated)
	Pyrotinib	PERMEATE [35]	No radiotherapy vs. radiotherapy	Phase II	---	36.0 months vs. 31.5 months	11.3 months (No radiotherapy) vs. 5.6 months (radiotherapy)
	Tucatinib	HER2CLIMB [36]	Tucatinib vs. Placebo added to trastuzumab and capecitabine	Phase III	---	21.9 months vs. 14.7 months	7.6 months (tucatinib) vs. 5.4 months (placebo plus trastuzumab and capecitabine)
ADC	T-DM1	EMILIA [27]	T-DM1 vs. Lapatinib plus capecitabine	Phase III	---	30.9 months vs. 25.1 months	9.6 months (T-DM1) vs. 6.4 months (lapatinib plus capecitabine)
	T-Dxd	TUXEDO-1 [37]	---	Phase II	---	---	14 months
		DESTINY-Breast03 [38]	T-Dxd vs. T-DM1	Phase III	---	---	25.1 months (T-Dxd) vs. 7.2 months (T-DM1)

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
