# Peer review of "Drug Treatment Direction Based on the Molecular Mechanism of Breast Cancer Brain Metastasis"

_pharmaceuticals, 2025, doi:10.3390/ph18020262_

Round 1
Reviewer 1 Report
Comments and Suggestions for Authors
Please find attached.

Author Response
Comment 1. Correct typos in abstract and throughout the manuscript
Response 1: Thank you for pointing this out. We have thoroughly reviewed the manuscript and corrected all typographical errors to ensure accuracy and clarity. This change can be found on line 20 and line 154.
Comment 2. Figure 1 not cited
Response 2: Thank you for pointing this out. We have already referenced figure 1 on line 48.
Comment 3. Lines 98-101 - the blood-brain barrier (BBB), a specialised structure made up of endothelial cells (ECs), pericytes, basement membranes, and astrocytes” is repeated, please correct.
Response 3: Thank you for pointing this out. As suggested, we have removed the duplicate part on Line 100 to ensure clarity and conciseness. We appreciate your careful attention to detail, which has helped improve the quality of our manuscript.
Comment 4. Table 1 is misaligned and the table and the following description neither match nor complement each other. Please harmonise the content of the table and the accompanying text, insert a reference in the text to Table 1, and complete the references for the studies listed in the table.
Response 4: Thank you for your comment regarding Table 1 and its alignment with the accompanying text. We have addressed the issues as follows:
Alignment and References:
We have corrected the alignment of Table 1 and ensured that all references for the studies listed in the table are complete and accurate.
Purpose of Table 1:
Table 1 summarizes targeted HER2 clinical trials for metastatic breast cancer patients, providing insights into potential treatment strategies for brain metastasis (BM) patients. Among these, only the PERMEATE and DESTINY-Breast03 trials are directed in patients with brain metastases, and these are explicitly referenced in the main text.
Complementary Information:
The discussions in the main text regarding tucatinib and T-DM1 are based on retrospective and exploratory analyses of brain metastasis subgroups from the HER2CLIMB and EMILIA trials, respectively. These analyses complement the data presented in Table 1, as they provide additional context and evidence for the efficacy of these therapies in BM patients.
Consistency with BM Focus:
The remaining clinical trials discussed in the main text are also focused on BM patients, ensuring that the content of the table and the text are harmonized and mutually supportive.
In addition, a new section on Neratinib was added in 3.1.2.2. A new experiment, TBCRC 022, is also added in Table 1.
We hope these revisions address your concerns and improve the clarity and coherence of the manuscript. Thank you again for your valuable feedback.
Comment 5. While the article provides an in-depth exploration of targeted therapies, the role of immune checkpoint inhibitors in BCBM is not addressed, which is a growing area in BCBM research. Please consider including a subsection on PD-1/PD-L1 inhibitors, as well.
Response 5: We sincerely appreciate your insightful suggestion regarding the inclusion of immune checkpoint inhibitors, particularly PD-1/PD-L1 inhibitors, in the context of breast cancer brain metastases (BCBM). In response to this comment, we have added a comprehensive subsection addressing the role of PD-1/PD-L1 inhibitors in BCBM, as detailed below:
Mechanism of PD-1/PD-L1 Axis in Immune Regulation
PD-1 is a type I transmembrane protein composed of 288 amino acid residues, belonging to the CD28 family. It is widely expressed on various immune cells, including activated T cells, B cells, NK cells, and dendritic cells (DCs). Structurally, PD-1 comprises an extracellular immunoglobulin variable (IgV) domain, a hydrophobic transmembrane domain, and a cytoplasmic tail domain containing immunoreceptor tyrosine-based inhibitory motifs (ITIMs) and immunoreceptor tyrosine-based switch motifs (ITSMs).
PD-L1, the ligand for PD-1, consists of 290 amino acid residues and is primarily composed of a short cytoplasmic tail region, a transmembrane region, and extracellular IgV and IgC-like domains. PD-L1 is expressed on tumor cells and antigen-presenting cells (APCs), including dendritic cells and macrophages. Additionally, PD-L1 expression can be induced on vascular endothelial cells in response to interferon-gamma (IFN-γ) stimulation. Upon binding of PD-1 on T cells to PD-L1 on tumor cells or APCs, the ITIMs and ITSMs of PD-1 are phosphorylated, leading to the recruitment and activation of the protein tyrosine phosphatase SHP-2 (Src homology region 2 domain-containing phosphatase 2). This process inhibits T-cell activation, promotes T-cell apoptosis, reduces cytokine production, and induces antigenic tolerance, thereby facilitating tumor immune evasion.
PD-L1 Expression and Immunogenicity in Breast Cancer
Breast cancer has traditionally been considered a non-immunogenic tumor. However, recent studies have revealed that PD-L1 expression and tumor-infiltrating lymphocytes (TILs) are significantly higher in triple-negative breast cancer (TNBC) patients compared to other breast cancer subtypes, suggesting that TNBC exhibits strong immunogenicity and may be amenable to anti-PD-1/PD-L1 therapy. Currently, pembrolizumab (a PD-1 inhibitor) and atezolizumab (a PD-L1 inhibitor) have been approved by the U.S. Food and Drug Administration (FDA) for the treatment of PD-L1-positive, unresectable, locally advanced, or metastatic TNBC.
Clinical Evidence for PD-1/PD-L1 Inhibitors in BCBM
Clinical trials such as KEYNOTE-355, KEYNOTE-522, and KEYNOTE-173 have demonstrated promising efficacy and manageable adverse effects of pembrolizumab in the neoadjuvant treatment of early-stage TNBC at high risk of recurrence, as well as in TNBC patients resistant to multiline chemotherapy regimens. However, these trials did not include patients with brain metastases. Emerging preclinical and clinical studies support the use of anti-PD-1/PD-L1 therapy in patients with brain metastases.
A preclinical study demonstrated that sequential ISIM treatment combined with anti-PD-L1 therapy exhibited synergistic antitumor efficacy and improved survival in a mouse model of breast cancer brain metastases. A phase II clinical trial involving 35 patients with breast cancer brain metastases (encompassing all four breast cancer subtypes) reported that 37% of patients achieved intracranial benefit from pembrolizumab treatment, with some experiencing mild adverse effects such as nausea, vomiting, and headache. These findings suggest that anti-PD-1/PD-L1 therapy holds promise for patients with brain metastases, although further trials are needed to validate these results.
Challenges and Future Directions
The exploration of neoadjuvant therapy for early-stage breast cancer in combination with immune checkpoint inhibitors remains limited but ongoing. For instance, an experimental approach combining the PD-L1 inhibitor atezolizumab with trastuzumab and pertuzumab was attempted in patients with HER2-positive breast cancer brain metastases. Unfortunately, no significant central nervous system (CNS) benefits were observed in these patients. Therefore, the potential benefits of PD-1/PD-L1 inhibitors for patients with brain metastases require further investigation through additional studies.
Detailed information on these therapies, including PD-1/PD-L1 inhibitors, can be found in Section 3.2.5. Programmed Cell Death Protein-1 (PD-1) / Programmed Cell Death Ligand 1 (PD-L1) of our revised manuscript.
We hope that this addition addresses your concern and provides a clearer understanding of the current landscape and potential of PD-1/PD-L1 inhibitors in the treatment of BCBM. We thank you for your valuable input, which has significantly strengthened the manuscript.
Comment 6. Lines 487-488 “metal nanoparticles and carbon nanoparticles can transport cisplatin to reduce its toxicity and enhance therapeutic efficacy” - The related citation is missing.
Response 6: Thank you for pointing this out. We have now included the reference to support the discussion on nanocarriers and their applications in drug delivery. Reference Added: Peña, Q.; Wang, A.; Zaremba, O.; Shi, Y.; Scheeren, H.W.; Metselaar, J.M.; Kiessling, F.; Pallares, R.M.; Wuttke, S.; Lammers, T. Metallodrugs in Cancer Nanomedicine. Chem. Soc. Rev. 2022, 51, 2544–2582. doi:10.1039/D1CS00468A.
Comment 7. Lines 489-490 “While liposomal nanocarriers can deliver doxorubicin and albumin nanoparticles in polymer nanoparticles can deliver paclitaxel” - This sentence is not entirely clear, please rephrase it.
Response 7: Thank you for pointing this out. Based on your suggestion, we have revised the sentence. The updated sentence now reads: "Liposomal nanocarriers can be used to deliver doxorubicin, while albumin-based nanoparticles within polymeric nanoparticles can be utilized for the delivery of paclitaxel."
Comment 8. Lines 47, 124, 343 and 442 - contain improper citations, displaying the error message 'Error! Reference source not found.'
Response 8: Thank you for pointing this out. We sincerely apologize for the improper citations and the error messages in the manuscript. We have carefully reviewed and corrected the citations on Lines 47, 124, 343, and 442 to ensure that all references are properly linked and displayed. All figures and tables mentioned in the text have been rechecked to ensure proper referencing and alignment with the manuscript content.
Reviewer 2 Report
Comments and Suggestions for Authors
Dear author and Editor,
The manuscript entitled "Drug treatment direction based on the molecular mechanism of 2 breast cancer brain metastasis: is well documented review about breast cancer brain metastases. However there are some major revisions needed before making it acceptable.
Comments
1) In the abstract Today, breast cancer (BC) represents the most commonly diagnosed malignancy among 10 the female population globally sentence seems incoherent as opening sentence frame. Author should reframe this statement.
2) In page no.1 line number 30 and 31 seems meaningless and proper reference citation needed for this report.
3) Page no.2 line number 46-47 is irrelevant statement with error notification. Author used AI tool for making such statements.
4) Page no.2 line no. 85 JAM-B can be hydrolyzed by tumor-derived histone S. is irrelavant. what menas Histone S?
5) In Brain metastases author did not explained about the role of Glioblastoma cells role in brain tumour. Author should provide detailed narrative about glioblastoma in brain cancer.
6) Author should provide information source details about the drugs used for HER2-targeting drugs they retreived.
7) Author should infer the limitations and side effects of the drug targeting the breast brain cancer metastases drug usages.
8) Manuscript contains many typographical errors like (Error! Reference source not found.) which is not acceptable.
Comments on the Quality of English Language
Overall English language of the manuscript is poor. Major language revision should be done using native English speaker.
Author Response
Comment 1. In the abstract Today, breast cancer (BC) represents the most commonly diagnosed malignancy among 10 the female population globally sentence seems incoherent as opening sentence frame. Author should reframe this statement.
Response 1: We sincerely thank you for pointing out the incoherence in the opening sentence. Based on your suggestion, we have reframed the sentence to improve its clarity and impact. The revised sentence now reads: "Breast cancer (BC) is the most frequently diagnosed malignancy and a leading cause of cancer-related deaths among women worldwide."
Comment 2. In page no.1 line number 30 and 31 seems meaningless and proper reference citation needed for this report.
Response 2: Thank you for your comment regarding the statement on Page 1, Lines 30–31. We have revised the text to remove data related to early-stage breast cancer, as it was not directly relevant to the focus of our study. In its place, we have cited the following reference to support the updated statement.
Reference:
Riecke, K.; Müller, V.; Neunhöffer, T.; Park-Simon, T.-W.; Weide, R.; Polasik, A.; Schmidt, M.; Puppe, J.; Mundhenke, C.; Lübbe, K.; et al. Long-Term Survival of Breast Cancer Patients with Brain Metastases: Subanalysis of the BMBC Registry. ESMO Open 2023, 8, 101213. doi:10.1016/j.esmoop.2023.101213.
This reference provides robust evidence on long-term survival outcomes in breast cancer patients with brain metastases, aligning with the focus of our study. We hope this revision addresses your concern and improves the clarity and relevance of the manuscript. Thank you again for your valuable feedback.
Comment 3. Page no.2 line number 46-47 is irrelevant statement with error notification. Author used AI tool for making such statements.
Response 3: We sincerely thank you for pointing out the irrelevant statement and error notification in Page 2, Lines 46-47. As suggested, we have removed the following sentence: "The extravasation of tumor cells into the brain also requires interaction with endothelial cells (ECs) on the BBB." This sentence was deemed unnecessary and inconsistent with the context of the section. We have carefully reviewed the manuscript to ensure that all remaining content is relevant, accurate, and free from errors.
Comment 4. Page no.2 line no. 85 JAM-B can be hydrolyzed by tumor-derived histone S. is irrelavant. what menas Histone S?
Response 4: We sincerely thank the reviewer for pointing out the irrelevant and unclear statement regarding "Histone S" in the manuscript. The phrase "JAM-B can be hydrolyzed by tumor-derived histone S" was indeed unclear and lacked proper context or definition. Upon review, we have determined that this statement is not relevant to the discussion and have removed it to maintain the clarity and scientific accuracy of the manuscript.
Comment 5. In Brain metastases author did not explained about the role of Glioblastoma cells role in brain tumour. Author should provide detailed narrative about glioblastoma in brain cancer.
Response 5: We sincerely thank you for your valuable feedback regarding the role of glioblastoma in brain tumors. We understand the importance of glioblastoma in the broader context of brain cancer research. However, as highlighted in our manuscript, this review focuses specifically on the mechanisms of breast cancer brain metastasis (BCBM) and aims to provide insights into existing and potential therapeutic strategies for patients with BCBM.
Glioblastoma, while a critical topic in brain cancer research, is not directly relevant to the scope of this review, which is centered on breast cancer metastasis to the brain. Including a detailed narrative on glioblastoma would deviate from the primary focus of the manuscript and could potentially dilute the clarity and relevance of the discussion for our intended audience.
We hope this clarification addresses your concern. If you feels that a brief mention of glioblastoma for comparative purposes would be beneficial, we would be happy to consider adding a short paragraph to contextualize the differences between primary brain tumors (e.g., glioblastoma) and metastatic brain tumors (e.g., BCBM).
BCCs infiltrate and grow to form distant metastases, whereas glioblastoma (GBM) originates from glial cell tissue, grows infiltratively into the brain parenchyma, and almost never metastasizes. However, both types of tumors must overcome the physical barriers provided by myoepithelial and astrocyte protrusions. In GBM, glioma cells are in direct contact with endothelial cells to destroy BBB. Additionally, in glioma, the existing vasculature is absorbed during initial tumorigenesis; however, subsequent tumor growth and hypoxia induce neoangiogenesis. This suggests the potential efficacy of VEGF inhibitors. For intracerebral tumors, it is crucial to use drugs that can cross the BBB. Preclinical studies have shown that temozolomide can distribute across the BBB, blood-tumor barrier (BTB), and cerebrospinal fluid (CSF) in mice. Currently, temozolomide is an important component of drug treatment for glioblastoma. However, its efficacy in breast cancer brain metastasis (BCBM) patients is limited[20], highlighting that different tumors exhibit varying BTB permeability.
Despite these differences, the overall treatment strategy remains the same: to deliver drugs across the BTB as effectively as possible to achieve therapeutic concentrations in the central nervous system (CNS).
In response to your comment, we have added a section at the end of "2. Mechanisms of brain metastasis" to address the similarities and differences between breast cancer brain metastasis (BCBM) and glioblastoma in terms of their interaction with the blood-brain barrier (BBB) and blood-tumor barrier (BTB). Thank you again for your thoughtful comments, which have helped us refine the focus and clarity of our manuscript.
Comment 6. Author should provide information source details about the drugs used for HER2-targeting drugs they retrieved.
Response 6: Thank you for your comment regarding the information sources for the HER2-targeting drugs discussed in our manuscript. We would like to clarify that all details about these drugs were retrieved from published literature, and we have now ensured that appropriate references are included to support the information provided. References have been added to Table 1 to ensure transparency and credibility. We hope this addresses your concern and provides the necessary context for the HER2-targeting drugs discussed in our work. Thank you again for your valuable feedback.
Comment 7. Author should infer the limitations and side effects of the drug targeting the breast brain cancer metastases drug usages.
Response 7: Thank you for pointing this out. We have added this part to the discussion section, and the revised discussion section is as follows:
This article details therapeutic strategies for BCBM, specifically drug therapy based on molecular mechanisms and delivery pathways across the BBB.
The molecular mechanisms underlying brain metastasis are intricate and encompass aberrant expression of numerous genes and signaling pathways. For instance, the HER2 gene is amplified, and the use of anti-HER2 drugs has been associated with enhanced patient survival in clinical settings. However, trastuzumab and pertuzumab are common with side effects such as cardiac dysfunction, rash, and fever. A trial of safety and tolerability in patients with HER2-positive breast cancer showed a higher incidence of diarrhoea and a higher incidence of neutropenia in the pertuzumab group than in the placebo group. TKI drugs (such as lapatinib, pyrotinib, tucatinib) may also cause diarrhea, abnormal liver function, rash, etc. The combination of a TKI with trastuzumab is more likely to cause diarrhea than trastuzumab alone. CDK4/6 inhibitors (such as abeciclib) may also cause neutropenia, leukopenia, anemia, etc. Long-term use may also cause adverse reactions such as fatigue, nausea, and vomiting.
ADC drug T-DXd may cause nausea, vomiting, fatigue, hair loss, etc. However, overall T-DM1 and T-DXd were manageable and safe for longer treatment durations. Common treatment-related adverse events with sacituzumab govitecan (SG) include neutropenia, leukopenia, and anemia. But it is well tolerated in patients with metastatic triple-negative breast cancer who have received significant prior therapy. In addition, PD-1/PD-L1 inhibitors may cause immune-related adverse events, such as immune pneumonia, immune hepatitis, and immune thyroid dysfunction.
Due to the complex mechanism of the signaling pathways and the involvement of multiple factors such as hypoxia in their activation, the precise role of activated Notch signaling in the brain-specific metastasis of tumor cells remains unclear. Preclinical trials related to this have shown the potential to inhibit tumor growth, but its role still lacks clinical trial data to confirm. In addition, brain metastases from breast cancer may acquire driver mutations after metastasis to the brain and carry mutations not detected in samples from the primary site. However, due to the small sample size, it is difficult to extract general evolutionary patterns between primary and metastatic foci, and biopsies of brain metastases in particular are more difficult. Currently, CSF interacting directly or indirectly with brain tumors may be a new direction for "liquid biopsies" based on brain tumor genetics.
The BBB restricts the entry of drugs into the CNS, such as trastuzumab and pertuzumab, due to their large molecular weight, making it difficult for them to cross the BBB. As a result, the concentration of these drugs in the CNS is significantly lower than in other parts of the body. Even if a drug successfully penetrates the BBB, tumor cells may develop resistance through multiple mechanisms, such as P-gp and LRP-1-mediated drug efflux. Therefore, our research is directed towards the development of enhanced drug delivery strategies that take into account the characteristics of this barrier. Examples include the utilization of RMT and the deployment of transporter protein inhibitors to augment the intracerebral concentration of the drug, thereby enhancing its therapeutic efficacy. Additionally, the article delineates various techniques for physically disrupting the BBB, as well as the utilization of nanocarriers (e.g., liposomes and gold nanoparticles) for the targeted delivery of drugs, ensuring efficient delivery to the tumor site while minimizing systemic toxicity. Despite diligent attempts to utilize a variety of nanocarrier, the anti-tumor efficacy of these carriers and their cumulative effect in the targeted tissues are still limited. To solve these challenges, various surface modifications (e.g., folic acid, histidine, transferrin, etc.) and stimulations (e.g., P and, enzymes, photothermal, etc.) have been applied to nanocarriers, but it is still a problem to accurately identify clinically applicable technologies from the wide variety of preclinical trials.
In addition, topical therapy is indispensable as an important complement to drug therapy. However, surgery is usually mainly applied to a single lesion and has limited effect on multiple brain metastases. WBRT may lead to irreversible cognitive impairment, with an increased risk of cognitive loss and a significant reduction in patients' quality of life with prolonged treatment. Although SRS has relatively few side effects, it is only suitable for brain metastases with 1~3 metastases and lesions ≤ 3~4cm in diameter, and it is still difficult to apply it multiple times.
Comment 8. Manuscript contains many typographical errors like (Error! Reference source not found.) which is not acceptable.
Response 8: Thank you for bringing the typographical errors in the manuscript to our attention. We sincerely apologize for the oversight regarding the incorrect references to figures and tables (e.g., "Error! Reference source not found"). We have carefully reviewed the manuscript and corrected all such errors. Proper citations for figures and tables have now been inserted, ensuring accuracy and consistency throughout the text.
Comment 9. Comments on the Quality of English Language: Overall English language of the manuscript is poor. Major language revision should be done using native English speaker.
Response 9: Thank you for your feedback regarding the quality of the English language in our manuscript. We sincerely apologize for any language-related issues that may have affected the clarity and readability of our work. To address this concern, we have thoroughly revised the manuscript with the assistance of professional language editing services. The revisions include improvements in grammar, syntax, sentence structure, and overall flow to ensure the manuscript meets the highest standards of academic writing. We hope these changes have significantly enhanced the quality of the language and made the manuscript more accessible to readers. Thank you again for your valuable feedback.
Reviewer 3 Report
Comments and Suggestions for Authors
The manuscript entitled "Drug treatment direction based on the molecular mechanism of breast cancer brain metastasis" is a study reporting the mechanistic pathways and molecular mechanisms involved in the progression of the tumor cells from the affected areas in the breast to the brain using the blood circulatory system and crossing the blood-brain barrier (BBB). The article also explains the invasion of these cells and their binding to the receptors as well as their afterwords repercussions. The theme of this survey is to report how the available anti-breast cancer drugs alter the molecular mechanisms and signaling pathways involved in the “breast cancer brain metastasis” and suggest the possible future therapeutic targets.
This study is useful in understanding the underlying mechanisms involved in breast cancer brain metastasis up to some extent. However, I have few major reservations that the authors are required respond and incorporate.
Comments are:
1. The article explicitly emphasizes on highlighting the general information about the main HER-2 targeted drugs but fails in giving insight view of the action of these drugs on their targets. The authors are encouraged to provide detailed schematic view of the mechanism of the action of the HER-2 anti-breast cancer drugs. This will be more interesting for the readers in understanding the underlying mechanisms involved and future direction in identifying such type of molecular targets.
2. A number of researchers have already reported such type of data in the recent past, for instance;
i. Ren, D., Cheng, H., Wang, X., Vishnoi, M., Teh, B. S., Rostomily, R., ... & Zhao, H. (2020). Emerging treatment strategies for breast cancer brain metastasis: from translational therapeutics to real-world experience. Therapeutic Advances in Medical Oncology, 12, 1758835920936151.
ii. Pedrosa, R. M., Mustafa, D. A., Soffietti, R., & Kros, J. M. (2018). Breast cancer brain metastasis: molecular mechanisms and directions for treatment. Neuro-oncology, 20(11), 1439-1449.
iii. Garrone, O., Ruatta, F., Rea, C. G., Denaro, N., Ghidini, M., Cauchi, C., ... & Rosenfeld, R. (2024). Current Evidence in the Systemic Treatment of Brain Metastases from Breast Cancer and Future Perspectives on New Drugs, Combinations and Administration Routes: A Narrative Review. Cancers, 16(24), 4164.
The authors are encouraged to justify the novelty of their work. How their work is different from the previously conducted studies on this topic.
3. Contrary to the text mentioned in the abstract, “This study of various drugs and techniques designed to boost the permeability of the BBB sheds light on how these innovations can improve the treatment of brain metastases” (Line-18-19; Page-1), we do not see any suggestions or conclusion drawn from the data shared how to improve the permeability of drugs across the blood brain barrier to intervene the metastases.
4. The article provides no detailed information about the non-HER-2 targeted anti-breast cancer drugs which is contrary to the title of this research.
5. It is not clear from this manuscript whether the authors have randomly taken few of the HER-2 targeted anti-breast cancer drugs to suggest future treatment directions or it’s all-time data as there is no time frame mentioned? The authors are encouraged to properly address this in the main body of the manuscript as well as in Abstract section to avoid such discrepancies.
6. The over all manuscript is highly disorganized because of the drug treatment direction part has been mixed with other sections. The authors are encouraged to put the suggested drug treatment direction part as a separate sub-heading at the end of each section, for the clear understanding of this whole study.
The authors are encouraged to properly address and incorporate these suggested major changes.
Author Response
Comment 1. The article explicitly emphasizes on highlighting the general information about the main HER-2 targeted drugs but fails in giving insight view of the action of these drugs on their targets. The authors are encouraged to provide detailed schematic view of the mechanism of the action of the HER-2 anti-breast cancer drugs. This will be more interesting for the readers in understanding the underlying mechanisms involved and future direction in identifying such type of molecular targets.
Response 1: Thank you for your valuable feedback. We appreciate your suggestion and have addressed this by adding a new Figure 2 to the manuscript, titled "Mechanism of Action of Anti-HER2 Drugs", which provides a comprehensive visual representation of the underlying molecular mechanisms. The HER family (HER1, HER2, HER3, HER4) plays a critical role in breast cancer, with HER2 overexpression being the most common. HER2 heterodimerizes with other EGFR family members, leading to autophosphorylation of tyrosine residues and activation of downstream signaling pathways such as Akt and MAPK, which promote cell proliferation and tumor growth.
As early as the 90s of the last century, the anti-HER2 monoclonal antibody trastuzumab has achieved good clinical efficacy, and then the new monoclonal antibody Pertuzumab inhibits the MAPK and PI3K signaling pathways by inhibiting HER2 dimerization and inhibiting the growth and development of tumor cells. With the development of antibody-drug conjugates, HER2 antibody conjugates T-DM1 and T-DXd have also achieved good efficacy, both of which enhance the killing effect on tumor cells by attaching small molecule toxin drugs to HER2 monoclonal antibodies and then transporting the toxin drug into the cell. Lapatinib, pyrotinib, and tucatinib, as TKIs, can also inhibit tumor development by inhibiting the MAPK signaling pathway.
The figure not only illustrates these mechanisms but also highlights the interplay between HER2 and its downstream signaling pathways, providing readers with a clearer understanding of how these drugs exert their therapeutic effects. Thank you again for your insightful suggestion, which has greatly improved the quality of our work. We hope the revised manuscript meets your expectations.
Comment 2. A number of researchers have already reported such type of data in the recent past, for instance; The authors are encouraged to justify the novelty of their work. How their work is different from the previously conducted studies on this topic.
Response 2: We sincerely appreciate your comment and the opportunity to clarify the novelty and distinct contributions of our work compared to previously conducted studies. We introduced the innovations in this article in the introduction section.
Many people have paid attention to these issues, but most of them regarding anti-HER2 treatment and local treatment. In addition to focusing on anti-HER2 treatment, this review focuses more on other potential targets and studies clinical drugs, such as CDK4/6 inhibitors. Trop-2 and PD-1/PD-L1, WNT and Notch pathway inhibitors are targeted. In addition, the role of BBB in brain metastasis in breast cancer is also discussed, and methods for treating BCBM are explained through BBB delivery drugs, such as physical pathways to increase BBB permeability, the use of nanotechnology for drug delivery, and drug delivery routes targeting BBB endothelial cells. The more detailed similarities and differences in each article are as follows.
- Ren, D., Cheng, H., Wang, X., Vishnoi, M., Teh, B. S., Rostomily, R., ... & Zhao, H. (2020). Emerging treatment strategies for breast cancer brain metastasis: from translational therapeutics to real-world experience. Therapeutic Advances in Medical Oncology, 12, 1758835920936151.
Targeting tumor–neural microenvironment interactions in BCBM, from basic experiments to study multiple signaling pathways between astrocytes and microglia. For instance, STAT3 inhibitors have achieved some efficacy in preclinical experiments, but they are still lacking Clinical data. The article also proposed that PDCH7 is a potential target, but all of them lack effective clinical data. The article summarizes local treatment. In addition, it also focuses on anti-HER2 treatment, but the data collected is less, pyrotinib is not mentioned, and PD-1/PD-L1 is not mentioned in immunotherapy. Other targeted drugs and BBB-related targets are not involved. The route of drug delivery across BBB using nano or physical technologies is not mentioned.
- Pedrosa, R. M., Mustafa, D. A., Soffietti, R., & Kros, J. M. (2018). Breast cancer brain metastasis: molecular mechanisms and directions for treatment. Neuro-oncology, 20(11), 1439-1449.
This review focuses on genetic predictors of brain metastases and looks for potential targets through sequencing, such as TP53, PIK3CA, KIT, MLH-1, and RB1. The review also mentions the mechanisms of brain metastases and the brain tumor microenvironment as well as potential targets such as STAT3. However, the clinical trials were not generalized, involving HER2, CDK4/6, etc., did not include PD-1/PD-L1, and did not discuss the drug delivery strategies across the BBB.
iii. Garrone, O., Ruatta, F., Rea, C. G., Denaro, N., Ghidini, M., Cauchi, C., ... & Rosenfeld, R. (2024). Current Evidence in the Systemic Treatment of Brain Metastases from Breast Cancer and Future Perspectives on New Drugs, Combinations and Administration Routes: A Narrative Review. Cancers, 16(24), 4164.
This review is relatively recent and explores a number of targets and clinical trials from the perspective of anti-HER2 therapy and TNBC therapy, which is roughly the same as our review target, and the research Strings and Inclusion/exclusion criteria are also introduced at the beginning of the article, but there is also no introduction to the trans-BBB drug delivery pathway.
Comment 3. Contrary to the text mentioned in the abstract, “This study of various drugs and techniques designed to boost the permeability of the BBB sheds light on how these innovations can improve the treatment of brain metastases” (Line-18-19; Page-1), we do not see any suggestions or conclusion drawn from the data shared how to improve the permeability of drugs across the blood brain barrier to intervene the metastases
Response 3: Thank you for your insightful comment regarding the statement in our abstract: “This study of various drugs and techniques designed to boost the permeability of the BBB sheds light on how these innovations can improve the treatment of brain metastases” (Lines 18-19, Page 1). We appreciate the opportunity to clarify how our study addresses the improvement of blood-brain barrier (BBB) permeability and its implications for treating brain metastases.
In Section "4. Therapeutic strategies based on BBB drug delivery pathways", we provide detailed discussions on several innovative approaches to enhance drug delivery across the BBB, as outlined below:
4.2. Physicochemical Pathway:
We discuss physical methods, such as focused ultrasound (FUS), which can disrupt tight junction proteins on BBB endothelial cells, thereby increasing BBB permeability. This technique allows drugs like carboplatin and paclitaxel to more effectively cross the BBB, enhancing their therapeutic potential for brain metastases.
4.1.1. LRP-1 Pathway:
We explore receptor-mediated transcytosis, specifically targeting the low-density lipoprotein receptor-related protein 1 (LRP-1) on BBB endothelial cells. For example, ANG1005, a peptide-chemotherapy conjugate, consists of three paclitaxel molecules covalently linked to Angiopep-2, a peptide designed to utilize the LRP-1 transport system. This design facilitates the transport of paclitaxel across the BBB, significantly improving its permeability and efficacy in treating brain metastases.
4.3. Application of Nanoparticles:
We highlight the unique physicochemical properties of nanoparticles, such as their small size, which enable them to act as effective drug carriers. Nanoparticles can encapsulate therapeutic agents, enhancing their solubility and BBB penetration. This approach improves the delivery of drugs across the BBB, offering a promising strategy for treating brain metastases.
These sections collectively demonstrate how our study provides concrete suggestions on improving BBB permeability to enhance drug delivery for brain metastases. We hope this clarification addresses your concern and underscores the relevance of our findings.
Thank you again for your valuable feedback.
Comment 4. The article provides no detailed information about the non-HER-2 targeted anti-breast cancer drugs which is contrary to the title of this research.
Response 4: Thank you for your comment regarding the need for more detailed information on non-HER2-targeted anti-breast cancer drugs. We appreciate your feedback and would like to clarify how our study addresses this aspect, which is indeed aligned with the title and scope of our research.
In our manuscript, we have included detailed discussions on several non-HER2-targeted therapies, as outlined below:
Section 3.2.3. CDK4/6 Pathway:
We discuss the efficacy of CDK4/6 inhibitors, such as abemaciclib, in HER2-negative breast cancer patients. Abemaciclib has shown promising results in this population, highlighting its role as a non-HER2-targeted therapy.
Section 3.2.4. Trophoblast Cell Surface Antigen-2 (Trop-2):
We explore the therapeutic potential of sacituzumab govitecan (SG), a Trop-2-targeting antibody-drug conjugate, which has demonstrated significant efficacy in treating metastatic triple-negative breast cancer (TNBC) patients. This represents another important non-HER2-targeted approach.
Additional Discussion on PD-1/PD-L1 Inhibitors:
We have also expanded our discussion to include PD-1/PD-L1 inhibitors, which have shown notable efficacy in TNBC patients. These immunotherapies further exemplify non-HER2-targeted strategies for breast cancer treatment.
Mechanism of PD-1/PD-L1 Axis in Immune Regulation
PD-1 is a type I transmembrane protein composed of 288 amino acid residues, belonging to the CD28 family. It is widely expressed on various immune cells, including activated T cells, B cells, NK cells, and dendritic cells (DCs). Structurally, PD-1 comprises an extracellular immunoglobulin variable (IgV) domain, a hydrophobic transmembrane domain, and a cytoplasmic tail domain containing immunoreceptor tyrosine-based inhibitory motifs (ITIMs) and immunoreceptor tyrosine-based switch motifs (ITSMs).
PD-L1, the ligand for PD-1, consists of 290 amino acid residues and is primarily composed of a short cytoplasmic tail region, a transmembrane region, and extracellular IgV and IgC-like domains. PD-L1 is expressed on tumor cells and antigen-presenting cells (APCs), including dendritic cells and macrophages. Additionally, PD-L1 expression can be induced on vascular endothelial cells in response to interferon-gamma (IFN-γ) stimulation. Upon binding of PD-1 on T cells to PD-L1 on tumor cells or APCs, the ITIMs and ITSMs of PD-1 are phosphorylated, leading to the recruitment and activation of the protein tyrosine phosphatase SHP-2 (Src homology region 2 domain-containing phosphatase 2). This process inhibits T-cell activation, promotes T-cell apoptosis, reduces cytokine production, and induces antigenic tolerance, thereby facilitating tumor immune evasion.
PD-L1 Expression and Immunogenicity in Breast Cancer
Breast cancer has traditionally been considered a non-immunogenic tumor. However, recent studies have revealed that PD-L1 expression and tumor-infiltrating lymphocytes (TILs) are significantly higher in triple-negative breast cancer (TNBC) patients compared to other breast cancer subtypes, suggesting that TNBC exhibits strong immunogenicity and may be amenable to anti-PD-1/PD-L1 therapy. Currently, pembrolizumab (a PD-1 inhibitor) and atezolizumab (a PD-L1 inhibitor) have been approved by the U.S. Food and Drug Administration (FDA) for the treatment of PD-L1-positive, unresectable, locally advanced, or metastatic TNBC.
Clinical Evidence for PD-1/PD-L1 Inhibitors in BCBM
Clinical trials such as KEYNOTE-355, KEYNOTE-522, and KEYNOTE-173 have demonstrated promising efficacy and manageable adverse effects of pembrolizumab in the neoadjuvant treatment of early-stage TNBC at high risk of recurrence, as well as in TNBC patients resistant to multiline chemotherapy regimens. However, these trials did not include patients with brain metastases. Emerging preclinical and clinical studies support the use of anti-PD-1/PD-L1 therapy in patients with brain metastases.
A preclinical study demonstrated that sequential ISIM treatment combined with anti-PD-L1 therapy exhibited synergistic antitumor efficacy and improved survival in a mouse model of breast cancer brain metastases. A phase II clinical trial involving 35 patients with breast cancer brain metastases (encompassing all four breast cancer subtypes) reported that 37% of patients achieved intracranial benefit from pembrolizumab treatment, with some experiencing mild adverse effects such as nausea, vomiting, and headache. These findings suggest that anti-PD-1/PD-L1 therapy holds promise for patients with brain metastases, although further trials are needed to validate these results.
Challenges and Future Directions
The exploration of neoadjuvant therapy for early-stage breast cancer in combination with immune checkpoint inhibitors remains limited but ongoing. For instance, an experimental approach combining the PD-L1 inhibitor atezolizumab with trastuzumab and pertuzumab was attempted in patients with HER2-positive breast cancer brain metastases. Unfortunately, no significant central nervous system (CNS) benefits were observed in these patients. Therefore, the potential benefits of PD-1/PD-L1 inhibitors for patients with brain metastases require further investigation through additional studies.
Detailed information on these therapies, including PD-1/PD-L1 inhibitors, can be found in Section 3.2.5. Programmed Cell Death Protein-1 (PD-1) / Programmed Cell Death Ligand 1 (PD-L1) of our revised manuscript.
These sections collectively provide substantial information on non-HER2-targeted therapies, including CDK4/6 inhibitors, Trop-2-targeting agents, and PD-1/PD-L1 inhibitors, all of which are highly relevant to HER2-negative and TNBC patients. We believe these discussions align with the title and scope of our research and address the need for detailed information on non-HER2-targeted drugs.
Thank you again for your valuable feedback. We hope this clarification addresses your concern.
Comment 5. It is not clear from this manuscript whether the authors have randomly taken few of the HER-2 targeted anti-breast cancer drugs to suggest future treatment directions or it’s all-time data as there is no time frame mentioned? The authors are encouraged to properly address this in the main body of the manuscript as well as in Abstract section to avoid such discrepancies
Response 5: Thank you for pointing this out. We appreciate your feedback and have addressed this concern in lines 175 to 184 of the manuscript.
According to Table 1, clinical trials for metastatic breast cancer have shown these drugs have good efficacy in patients with metastatic HER2-positive breast cancer and can therefore serve as the treatment direction for patients with brain metastasis.
In addition, according to clinical guidelines (e.g., NCCN guidelines), trastuzumab and pertuzumab combined with chemotherapy are the first-line treatment options for the treatment of HER2-positive breast cancer patients. Therefore, trastuzumab and pertuzumab are preferred for patients with brain metastasis. However, patients with advanced brain metastasis often have drug resistance to first-line medications, so other treatment options can be tried, such as ADC and TKI drugs.
Currently, the FDA-approved ADC drugs targeting HER2 for breast cancer treatment include T-DM1 (FDA approved T-DM1 in 2013 for the treatment of HER-2-positive MBC patients who have received trastuzumab and first-line taxane chemotherapy.) and T-DXd (In 2019, the FDA accelerated approval of T-DXd for patients with HER2-positive advanced breast cancer who have previously received ≥2-line anti-HER2 treatment in the metastatic stage), TKI drugs include lapatinib (in 2007), neratinib (2017), and tucatinib (2020). Pyrotinib has not been approved, but clinical trials have also shown good efficacy.
Comment 6. The over all manuscript is highly disorganized because of the drug treatment direction part has been mixed with other sections. The authors are encouraged to put the suggested drug treatment direction part as a separate sub-heading at the end of each section, for the clear understanding of this whole study.
Response 6: Thank you for pointing this out. We appreciate your feedback and have added a new section to the manuscript in 3.2.6. In summary.
The treatment landscape for breast cancer brain metastases (BCBM) is evolving rapidly, with significant advancements in targeted therapies and a deeper understanding of molecular mechanisms. The following points summarize the current status and future directions for therapeutic strategies. Nowadays, anti-HER2 treatments have demonstrated substantial clinical benefits in patients with HER2-positive BCBM. Drugs such as trastuzumab, pertuzumab, lapatinib, neratinib, tucatinib, T-DM1, and T-DXd have shown significant efficacy in prolonging progression-free survival (PFS) and overall survival (OS) in clinical trials. The use of anti-HER2 therapies is well-established and recommended for patients with HER2-positive BCBM. Additionally, WNT and Notch signaling pathways play a crucial role in tumor biology and metastasis. However, their mechanisms are highly complex and diverse. Current research on WNT and Notch inhibitors is primarily at the preclinical stage. While these pathways show promise as therapeutic targets, further studies are needed to elucidate their roles and develop effective inhibitors for clinical use. In addition, the CDK4/6 inhibitor abemaciclib has shown promising results in HR+/HER2- BCBM patients. With its ability to cross the blood-brain barrier (BBB) and achieve effective concentrations in the brain, abemaciclib represents a new direction for the treatment of BCBM. Ongoing clinical trials are further exploring its potential in this patient population. Sacituzumab govitecan (SG), trop-2-targeted ADC, has demonstrated significant efficacy in preclinical models and clinical trials with small sample sizes. The promising results in both mice and patients suggest that SG could be a valuable treatment option for BCBM. Larger-scale clinical trials are warranted to confirm its efficacy and safety in a broader patient population. Anti-PD-1/PD-L1 therapies have achieved remarkable success in non-brain metastatic triple-negative breast cancer (TNBC) patients. However, data on their efficacy in patients with brain metastases are limited. Preclinical studies and early clinical trials have shown some potential benefits, but more research is needed to determine their role in the treatment of BCBM.
In conclusion, while significant progress has been made in the treatment of BCBM, there are still many challenges to overcome. Future research should focus on developing more effective targeted therapies.
Round 2
Reviewer 2 Report
Comments and Suggestions for Authors
Dear Author,
The manuscript is revised well and all the mentioned review comments were carried out perfectly. I agree with manuscript in the present form.
Reviewer 3 Report
Comments and Suggestions for Authors
The revised version of the manuscript entitled "Drug treatment direction based on the molecular mechanism of breast cancer brain metastasis" is in much better condition now. The authors have made major changes to the manuscript. The authors have positively responded to all the concern shown apart from comment No. 2 relating to the novelty of this work. The work done by Garrone et. al. [Garrone, O., Ruatta, F., Rea, C. G., Denaro, N., Ghidini, M., Cauchi, C. & Rosenfeld, R. (2024). Current Evidence in the Systemic Treatment of Brain Metastases from Breast Cancer and Future Perspectives on New Drugs, Combinations and
Administration Routes: A Narrative Review. Cancers, 16(24), 4164] is closely related to this work but not exactly the same. Here the response of the authors that the mentioned article is not furnishing details of trans-BBB drug delivery pathway contrary to their own manuscript seems partially logical and can be accepted. Moreover, since this over all study is interesting and will be an eye catcher for the readers. Therefore, I accept this article for publication in this journal in this current form.